# Method development and characterisation of the low-molecular-weight peptidome of human wound fluids

Mariena JA van der Plas[1,2]*, Jun Cai[2†], Jitka Petrlova[1†], Karim Saleh[1,3], Sven Kjellström[4], Artur Schmidtchen[1,3,5]

[1]Division of Dermatology and Venereology, Department of Clinical Sciences, Lund University, Lund, Sweden; [2]LEO Foundation Center for Cutaneous Drug Delivery, Department of Pharmacy, University of Copenhagen, Copenhagen, Denmark; [3]Dermatology, Skane University Hospital, Lund, Sweden; [4]Division of Mass Spectrometry, Department of Clinical Sciences, Lund University, Lund, Sweden; [5]Copenhagen Wound Healing Center, Bispebjerg Hospital, Department of Biomedical Sciences, University of Copenhagen, Copenhagen, Denmark

**Abstract** The normal wound healing process is characterised by proteolytic events, whereas infection results in dysfunctional activations by endogenous and bacterial proteases. Peptides, downstream reporters of these proteolytic actions, could therefore serve as a promising tool for diagnosis of wounds. Using mass-spectrometry analyses, we here for the first time characterise the peptidome of human wound fluids. Sterile post-surgical wound fluids were found to contain a high degree of peptides in comparison to human plasma. Analyses of the peptidome from uninfected healing wounds and *Staphylococcus aureus* -infected wounds identify unique peptide patterns of various proteins, including coagulation and complement factors, proteases, and antiproteinases. Together, the work defines a workflow for analysis of peptides derived from wound fluids and demonstrates a proof-of-concept that such fluids can be used for analysis of qualitative differences of peptide patterns from larger patient cohorts, providing potential biomarkers for wound healing and infection.

*For correspondence:
mariena.van_der_plas@med.lu.se

†These authors contributed equally to this work

Competing interests: The authors declare that no competing interests exist.

## Introduction

Wound infections after surgery and in relation to burns, as well as in non-healing wounds, are significant medical and societal problems (*Sen et al., 2009*). Surgical site infections (SSIs) are leading nosocomial infections in developing countries and the second most frequent nosocomial infections in Europe and the United States (*Allegranzi et al., 2011*). For example, in European hospitals, the overall rates of SSI range between 3% and 4% of patients undergoing surgery (*Saleh and Schmidtchen, 2015*). In some procedures, such as when using skin grafts, or performing hernia surgery using biomaterials, the infection risk is higher and may exceed 5–10% (*Saleh and Schmidtchen, 2015*; *Futoryan and Grande, 1995*; *Falagas and Kasiakou, 2005*). The economic burden of failing skin repair is therefore extensive (*Lindholm and Searle, 2016*). Currently, the costs of treating wounds are estimated to be over 3% of the total health care budgets of Western countries (*Guest et al., 2017*; *Phillips et al., 2016*) and these expenses are projected to grow with the increasing development of antimicrobial resistance, as well as ageing of the population and the rising incidence and prevalence of diseases such as obesity and diabetes (*MedMarket Diligence, 2021*). All these factors contribute not only to an increased risk for SSI, but also to an increase in the incidence of non-healing wounds in elderly patients with circulatory insufficiencies. Besides the enormous economic burden, wound infections in acute and non-healing wounds lead to increased risk of

**eLife digest** Infected wounds and burns represent a serious risk to patients: they can delay healing and, if left untreated, can lead to generalised infection or sepsis, organ failure and death. Wounds and burns get infected when harmful micro-organisms, such as bacteria, enter the wound. Predicting the risk of infections, and detecting them early, could reduce their impact and make treating them easier.

A way to distinguish between healing and infected wounds is to study how proteins are broken down in each situation. Proteases are the enzymes that break down proteins, and they are different in healing wounds and infected wounds that are failing to heal. This is because, while the body produces proteases, the bacteria that cause infection do so too. Each protease breaks down proteins in a specific way, resulting in a different set of protein fragments, known as peptides. Together, all the peptides in a wound are referred to as the wound's 'peptidome'. Studying the peptidome of a wound could show whether it is infected, and even what type of bacteria might be responsible, which could help identify suitable treatments.

Van der Plas et al. used a technique called mass spectrometry to study the peptidome of wounds after surgery. Sterile post-surgical wounds showed high levels of peptides compared to plasma, the liquid component of blood, with up to 4,300 different peptides. Comparing healing wounds to ones infected with the bacterium *Staphylococcus aureus* revealed that infected wounds contained peptides from about 150 proteins not found in uninfected wounds, while peptides from 90 proteins were unique to uninfected wounds. The peptides exclusive to uninfected wounds included some linked to antimicrobial activity and immune system activity.

Van der Plas et al.'s results suggest that analysing the peptidome may be an approach to tracking the healing status of wounds, making it easier to detect infection before symptoms are apparent. The next step will be to study more wounds and identify the reliable peptide markers to use them for diagnostic tests.

invasive infections and sepsis, dysfunctional wound closure, and risk for unaesthetic scarring, as well as a reduced quality of life (*Consensus I, 2012*).

Given the above, there is an unmet need for improved methods to measure and predict wound healing status and infection risk in different types of wounds. Furthermore, detailed characterisation of wounds may enhance our understanding of the causes that lead to failure and possibly reveal novel therapeutic targets. A non-invasive way of investigating wound healing is through analysis of wound exudates. These contain proteins, peptides and other biological components, such as metabolites, which can be characterised to various degrees. In agreement, several studies have reported the use of proteomics for the analysis of wound fluids (*Eming et al., 2010*; *Kalkhof et al., 2014*; *Escalante et al., 2009*). However, although a powerful methodology, mass spectrometry analyses on samples after trypsin digestion of the proteins mainly report on the presence of proteins and their high-molecular-weight fragments, whereas information about the endogenous fragmentation and resulting peptides is lost. In wounds, however, endogenous protein degradation is of high relevance for the understanding of healing as balanced proteolytic activity is essential for progression of healing, whereas aberrant protease activity, such as seen in patients with infected acute wounds (*Trengove et al., 1999*; *Saleh et al., 2019*) and non-healing ulcers (*Trengove et al., 1999*; *Grinnell and Zhu, 1996*), may lead to deteriorated wound healing. Various endogenous proteases, such as neutrophil elastase, matrix metalloproteases, and collagenases, play important roles in both functional and impaired healing (*Trengove et al., 1999*; *Agren, 2000*; *Yager and Nwomeh, 1999*; *Herrick et al., 1997*). Additionally, exogenous proteases secreted by colonising or infecting bacteria may influence healing as well (*McCarty and Percival, 2013*; *Suleman, 2016*). These different proteases may degrade endogenous proteins into peptides with sequences that are enzyme specific, and this may result in peptide patterns that reflect the nature and level of protein degradation occurring in a wound. This subject has been addressed to some extend by Sabino et al. who applied quantitative proteomics strategies to assess the wound proteome and the activity of distinct protease groups along the healing process, thus mapping proteolytic pathways (*Sabino et al., 2018*; *Sabino et al., 2015*). However, given the dynamics of wound healing and the highly proteolytic

environment, the generation of endogenous peptides in wounds, their structures, and possible utilisation as biomarkers for wound healing still remains to be explored.

The large-scale analysis of endogenous peptides has, since its introduction in 2001 (*Schrader and Schulz-Knappe, 2001*; *Bergquist and Ekman, 2001*; *Schulz-Knappe et al., 2001*), resulted in a new omics field, that is peptidomics or peptidome research. Peptidomics investigations have been conducted for a number of different biological samples, including plasma (*Tammen et al., 2005*), cerebrospinal fluid (*Hölttä et al., 2012*), saliva (*Vitorino et al., 2012*), tears (*Hayakawa et al., 2013*), and brain tissues (*Secher et al., 2016*). As the low-molecular-weight peptidome of wounds could act as downstream reporters of the proteolytic action of endogenous and exogenous proteases they could serve as a promising tool for diagnosis and/or prognosis of wound healing. In a recent study, we showed that peptides from a selected protein, human thrombin, are detected and could be attributed to proteolytic actions (*Saravanan et al., 2017*). Specific thrombin-derived peptide sequences were identified in wound fluids from acute and non-healing ulcers, respectively. The result, although focusing on one single protein, demonstrated a proof-of-concept pointing at the possibility of defining peptide biomarkers for improved diagnosis of wound healing and infection. In the present study, we aimed at developing a robust peptidomics method for the characterisation of the peptidome of wound fluids. Using this method, we here compare acute non-infected wound fluids with plasma samples and find significantly higher protein and peptide numbers in wound fluids compared with plasma, which typically were also smaller in size as compared to plasma-derived peptides. Finally, we analyse wound fluids collected from dressings after facial surgery and compare three uninfected and normally healing surgical wounds with three inflamed and *Staphylococcus aureus* -infected wounds. We further demonstrate the utility of peptidomics in wound fluid analysis, showing peptide profiles of various selected proteins. Together, the work defines a workflow for analysis of peptides derived from human wound fluids and demonstrate a proof-of-concept that such wound fluids can be used for analysis of subtle qualitative differences in peptide patterns derived from individual patient samples. Moreover, as recently also demonstrated (*Hartman et al., 2021*), the uploaded data sets derived from the present report were further mined and processed using global bioinformatic approaches exploring quantitative differences in the identified peptidomes, demonstrating the applicability of the peptidome data generated in this study.

## Results

### Comparison of sample preparation methods

To determine the most optimal method for extraction of peptides from wound fluids, acute wound fluids were mixed with 6 M (final concentration) urea in the absence or presence of 0.05% *Rapi*Gest or 0.1% TFA followed by 30 kDa filtration as indicated in *Figure 1A*. SDS–PAGE analysis of the filtrate showed high-molecular-weight proteins before filtration (BF) and low-molecular-weight peptides after filtration (*Figure 1B*). Urea alone (U) did only result in a few peptide bands, whereas both *Rapi*Gest (U + R) and TFA (U + T) resulted in more abundant peptide patterns. To further analyse and optimise peptide extraction, various volumes of wound fluids were filtrated as described above, followed by desalting and concentrating of defrosted samples using StageTips and finally liquid chromatography–tandem mass spectrometry (LC–MS/MS) analysis of peptides between 700 and 6400 dalton (the workflow is depicted in *Figure 1A*) was performed. The results showed a sample volume-dependent increase in the number of identified peptides and corresponding proteins for all used buffers (*Figure 1C*). In agreement with the SDS–PAGE results, urea alone resulted in lower numbers of identified proteins and unique peptides than the other two buffers. Interestingly, similar numbers of proteins were identified when using urea and TFA or urea and *Rapi*Gest (180 versus 173) for 100 μL of WF, whereas more unique peptides were found using the latter buffer (2931 versus 3543). Based on these results, 100 μL of wound fluids in 6 M urea supplemented with 0.05% *Rapi*Gest was selected for further studies.

### Robustness of sample preparation method

To determine the robustness of the selected sample preparation method, each step of the workflow was investigated. The results showed no difference in peptide patterns on SDS–PAGE after dividing one sample preparation over two 30 kDa cut-off filters (*Figure 2A*). Nevertheless, LC–MS/MS

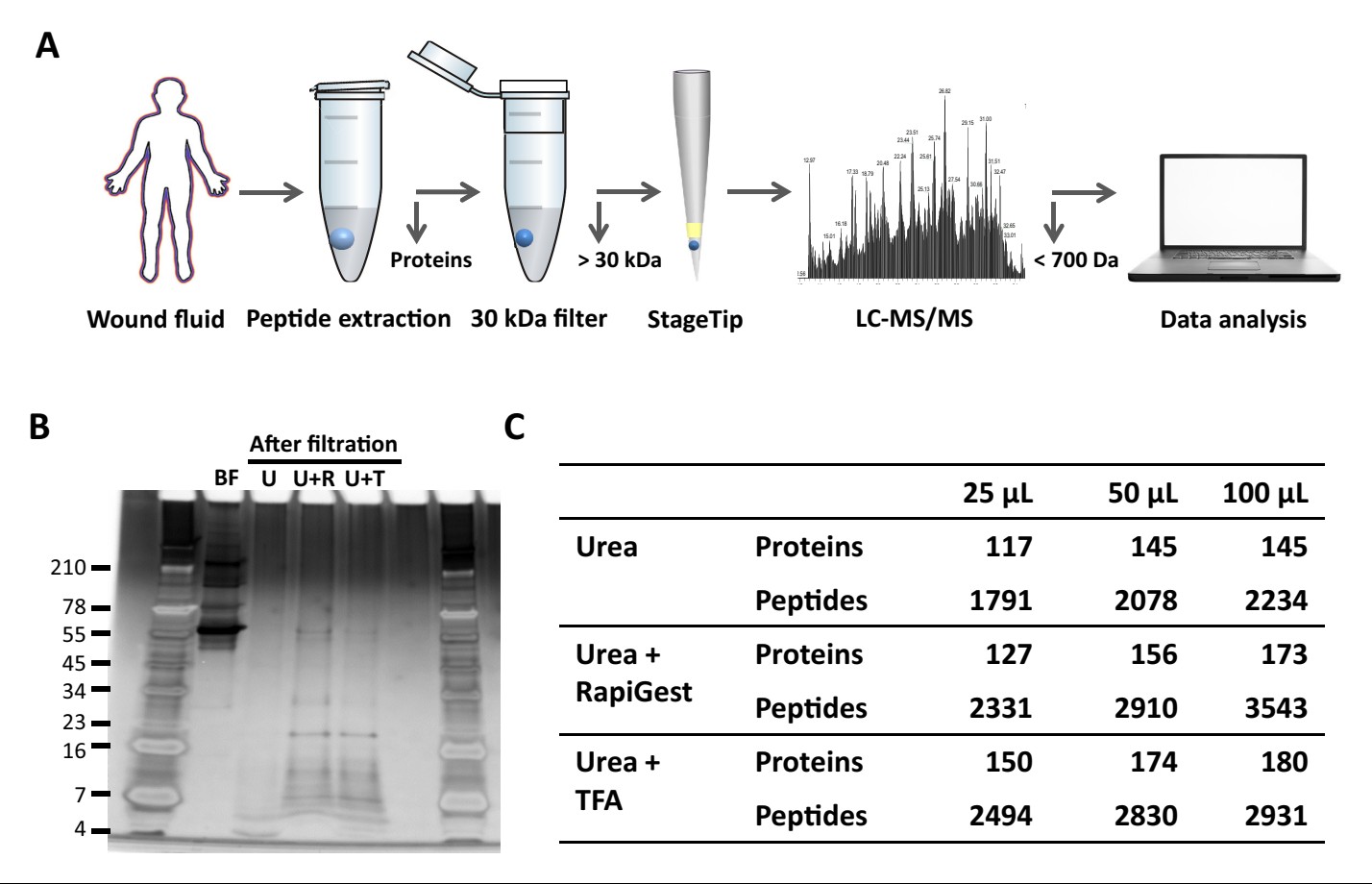

**Figure 1.** Comparison of sample preparation methods. (**A**) Schematic overview of the workflow. Peptides were extracted from 25 µL, 50 µL, or 100 µL of wound fluid in 6 M urea (U), 6 M urea +0.05% *Rapi*Gest (U + R), or 6M urea +0.1% TFA (U + T), using 30 kDa cut-off filters. Stored filtrates were defrosted, followed by peptide concentration using StageTips and finally 1.6 µL of the original wound fluids were analysed by nano-LC–MS/MS. (**B**) Representative example of a 10–20% Tricine gel run with 1 µL of sample before filtration (BF) or 22.5 µL of sample after filtration, extracted from 100 µL of wound fluid, ran under non-reducing conditions, and stained with SilverQuest stain. (**C**) Total numbers of identified peptides and corresponding proteins for the different buffers and amounts of wound fluid as analysed by MS. Results are shown as combined data of two injections per sample.

analyses of two injections per sample showed that 62% (170) of the peptides came from the same proteins (Venn diagrams, *Figure 2A*), whereas 58% (3363) of all unique peptides were found in both samples, indicating that the filters do interfere with peptide recovery. Next, we investigated the reproducibility of sample injection using one preparation injected three times on the same day. As shown in *Figure 2B*, we found similar heatmap patterns for the protein scores for the three injections. Furthermore, similar patterns were also found for the number of unique peptides identified and the percentage coverage of the protein by these peptides. Moreover, comparison of one preparation injected twice on two different days resulted in clear correlations in protein score, number of unique peptides, and protein coverage (*Figure 2C*), indicating that sample storage at 4°C does not significantly influence the results. Finally, we tested the reproducibility of the entire sample preparation method by comparing two independently generated samples of the same wound fluid and found a good correlation for all three measured parameters (*Figure 2D*). Taken together, the above results show that the selected sample preparation method and storage is robust and can be used for peptidome comparison of various donor fluids.

## Comparison of plasma and wound fluids

Whereas proteases are activated and/or released in the wound environment during inflammation, plasma from healthy donors should not contain activated proteases and therefore far less peptides.

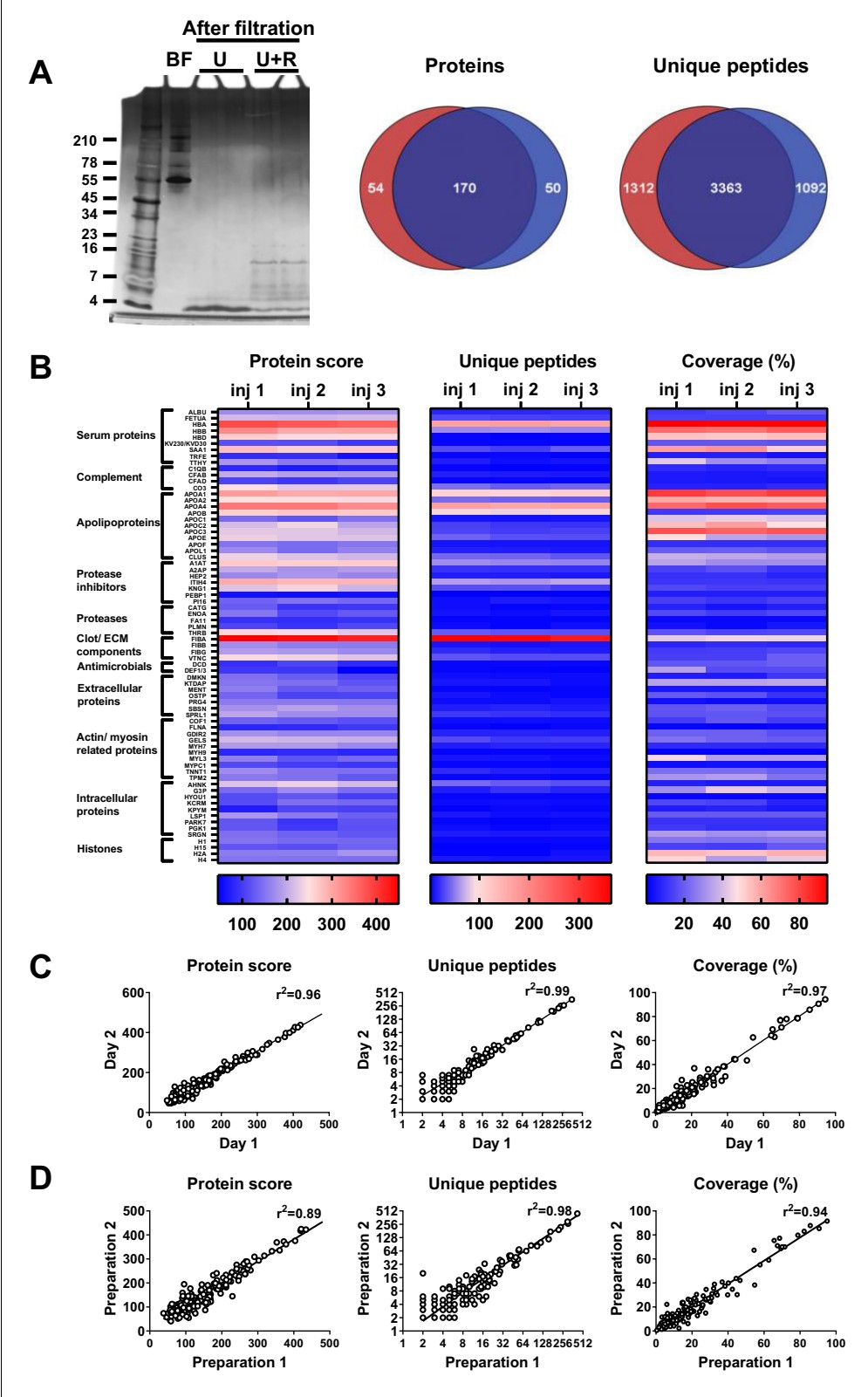

**Figure 2.** Robustness of sample preparation method. Peptides were extracted from 100 μL wound fluid in 6 M urea supplemented with 0.05% *RapiGest* using the workflow shown in *Figure 1*. (**A**) To investigate possible influence of the used filters on the obtained results, one wound fluid preparation was divided over two 30 kDa cut-off filters, centrifuged and filtrates were analysed using SDS–PAGE (U + R) and nano-LC–MS/MS. Combined results from two injections per sample are shown in Venn diagrams depicting proteins and unique peptides. For SDS–PAGE, 1 μL of sample before filtration (BF) or

*Figure 2 continued on next page*

*Figure 2 continued*

22.5 µL of sample after filtration were run on a 10–20% Tricine gel under non-reducing conditions and stained with SilverQuest stain; extractions using 6M urea (U) without *Rapi*Gest are shown for comparison. (B) Reproducibility of sample injection using one preparation injected three times on the same day or (C) injected twice on two different days. (D) Reproducibility of sample preparation using two independently generated samples of the same wound fluid on different days. Combined results of two injections per sample are expressed as the protein score, the number of unique peptides per protein, and the percentage of total coverage of each protein by the identified peptides; $r^2$ values are indicated in each graph.

Indeed, a clear difference between three wound fluids and three plasma samples could be observed on SDS–PAGE, the latter samples containing fewer distinct bands and of a higher molecular weight (*Figure 3A*). In agreement, LC–MS/MS analysis showed substantially more peptides in wound fluids as compared to plasma (*Figure 3B*). When pooling the results of the three individuals for each fluid type, over five times as many proteins and 6.8 times as many peptides were observed in wound fluids as compared to plasma samples (*Figure 3C*). Moreover, only 9.6% (30) of all proteins and 2% (121) of all peptides were detected in both types of fluids, although not necessarily in all three fluids in each group. Interestingly, we found relatively more small peptides (700–1500 Da) in wound fluids, whereas peptides larger than 1700 Da were relatively more prevalent in plasma (*Figure 3D*, right panel). Finally, heatmaps of the proteins that were common either for the three plasma samples or for the three wound fluid samples were generated, ordered on function or localisation, showing clear differences between these two types of samples in the protein score, number of unique peptides, and coverage (*Figure 3E*). As expected, no peptides derived from proteases were detected in plasma. Besides the number of identified peptides, another difference between plasma and wound fluids is the degree and type of post-translational modifications (a summary of mass spectrometry results for all sample types is shown in *Supplementary file 1*). For each wound fluid, we found a higher degree of peptide modifications than in plasma. In plasma, the average deamidation of the peptides was 3.1% as compared to 6.4% for aWF sample, whereas the degree of oxidation of methionine was 21.7% in wound fluids, while only 4.5% in CP samples. All peptides identified in the plasma and wound fluid samples are listed in Supplementary Datasets 1 and 2.

## Comparison of acute wound fluids

To investigate similarities of sterile acute wound fluids, five wound fluids from different donors were processed and analysed with LC–MS/MS. The data of four injections per sample (two injections a day at two different days) were merged and then subjected to a database search. For all wound fluids, high numbers of peptides were detected (ranging from 2649 to 4271, *Supplementary file 2*) and these peptides corresponds to a total of 373 proteins (*Supplementary file 1*). As illustrated in *Figure 4A*, 74 proteins were detected in all five wound fluid samples. Within this group of 74 proteins, there were 783 identical peptides (*Figure 4A*, second Venn diagram), which amounts to 15% of the total peptides for the common proteins. Notably, the slightly higher number of common peptides in the Venn diagram for all detected proteins (*Figure 4A*, third Venn diagram) as compared to that found for the common proteins can be explained by the presence of 14 peptides that were not unique for one of the 74 common proteins, but were also assigned to other proteins that were not included in the common group of proteins. Interestingly, we found a difference in the average peptide length, ranging from 12.3 to 13.7 amino acids (AA), of the identified peptides from the five different wound fluids (*Supplementary file 2*), suggesting that the samples have been exposed to different levels of proteolytic activity. Finally, heatmaps of the proteins that were common for all five samples were generated showing resemblances in protein score, numbers of unique peptides, and protein coverage (*Figure 4B*), which further proves the usefulness of peptidomics in wound fluid analysis.

## Comparison of non-inflamed, non-infected wounds with inflamed and infected wounds

Finally, and as a proof-of-concept, we selected six patients from a previously published clinical study (*Saleh et al., 2019*), who had undergone facial full-thickness skin grafting, and either had healed well at the 7 day follow-up after surgery (*Figure 5A*, low inflammation group) or had an inflamed wound infected by amongst others *S. aureus* (high inflammation group). Cytokine analysis of extracts made of the wound dressings indeed showed increased levels of IL-1β, IL-6, IL-8, and TNF-α in the

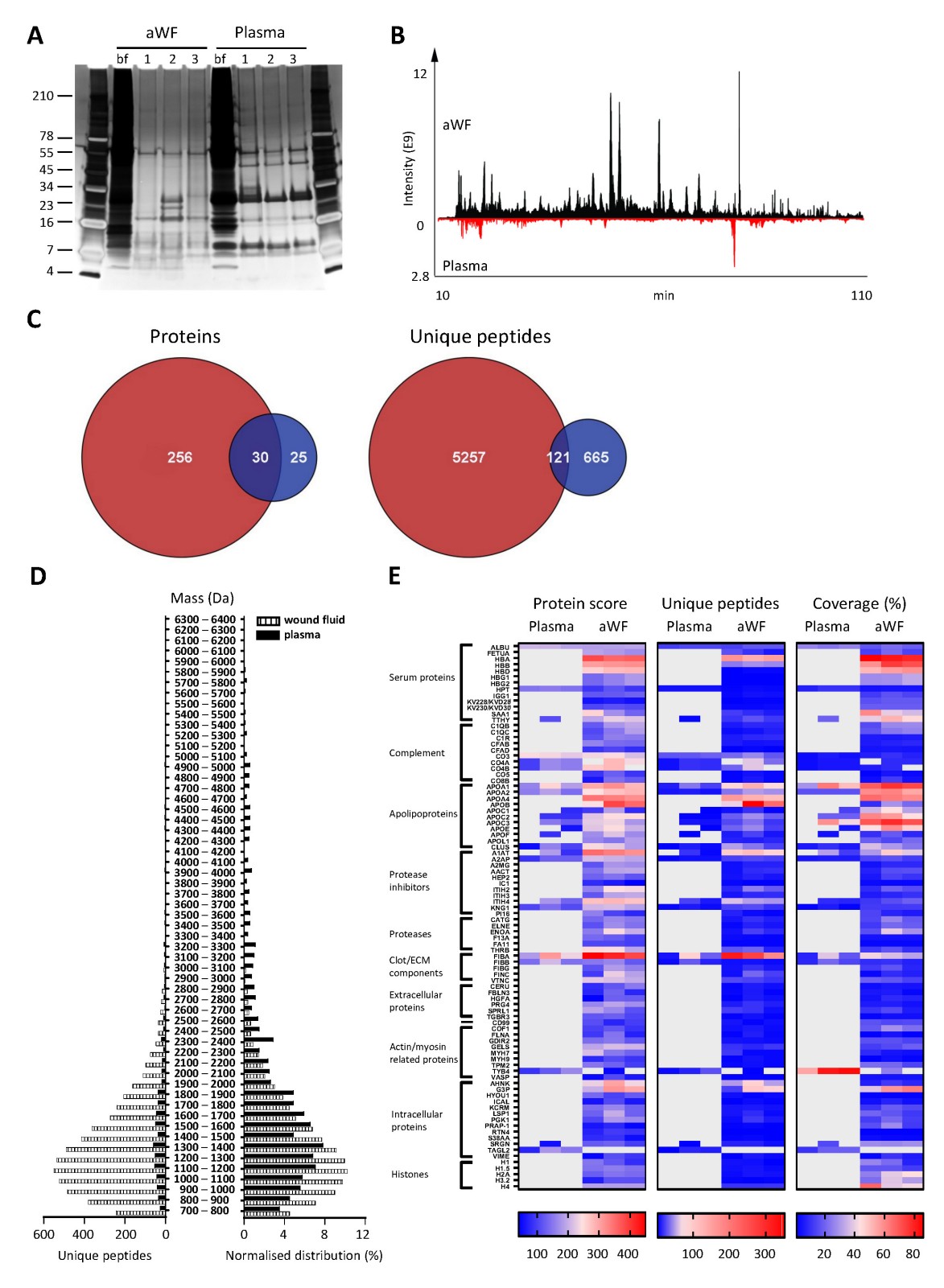

**Figure 3.** Comparison of plasma and wound fluids. Peptides were extracted from 100 μL acute wound fluid (aWF) or citrated plasma in 6 M urea supplemented with 0.05% *Rapi*Gest using the workflow shown in *Figure 1*. (**A**) Comparison of three wound fluids and three plasma samples as analysed using a 10–20% Tricine gel ran under non-reducing conditions and stained with SilverQuest stain. (**B**) Representative LC–MS/MS chromatograms of wound fluid (top) and plasma (bottom) preparations. (**C**) Comparison of the pooled results of three wound fluids with three plasma preparations using

*Figure 3 continued*

Venn diagrams depicting total number of identified proteins and unique peptides. (D) Distribution of peptides from representative wound fluid and plasma preparations based on molecular weight. The results are shown as total numbers (left) and normalised values (right). (E) Heatmaps comparing wound fluids and plasma depicting the protein score, the number of unique peptides per protein, and the percentage of total coverage of each protein by the identified peptides. Results are shown as combined data of two injections per sample.

high inflammation as compared to the low inflammation group (*Figure 5B*). Further analyses of the dressing extracts using SDS–PAGE (*Figure 5C*) and zymograms (*Figure 5D*) showed a positive correlation between protein degradation and enzymatic activity, both more apparent in the high inflammation group. Next, two independently generated sample preparations were made of each extract, using the sample preparation method above, and subjected to LC–MS/MS. The data of four

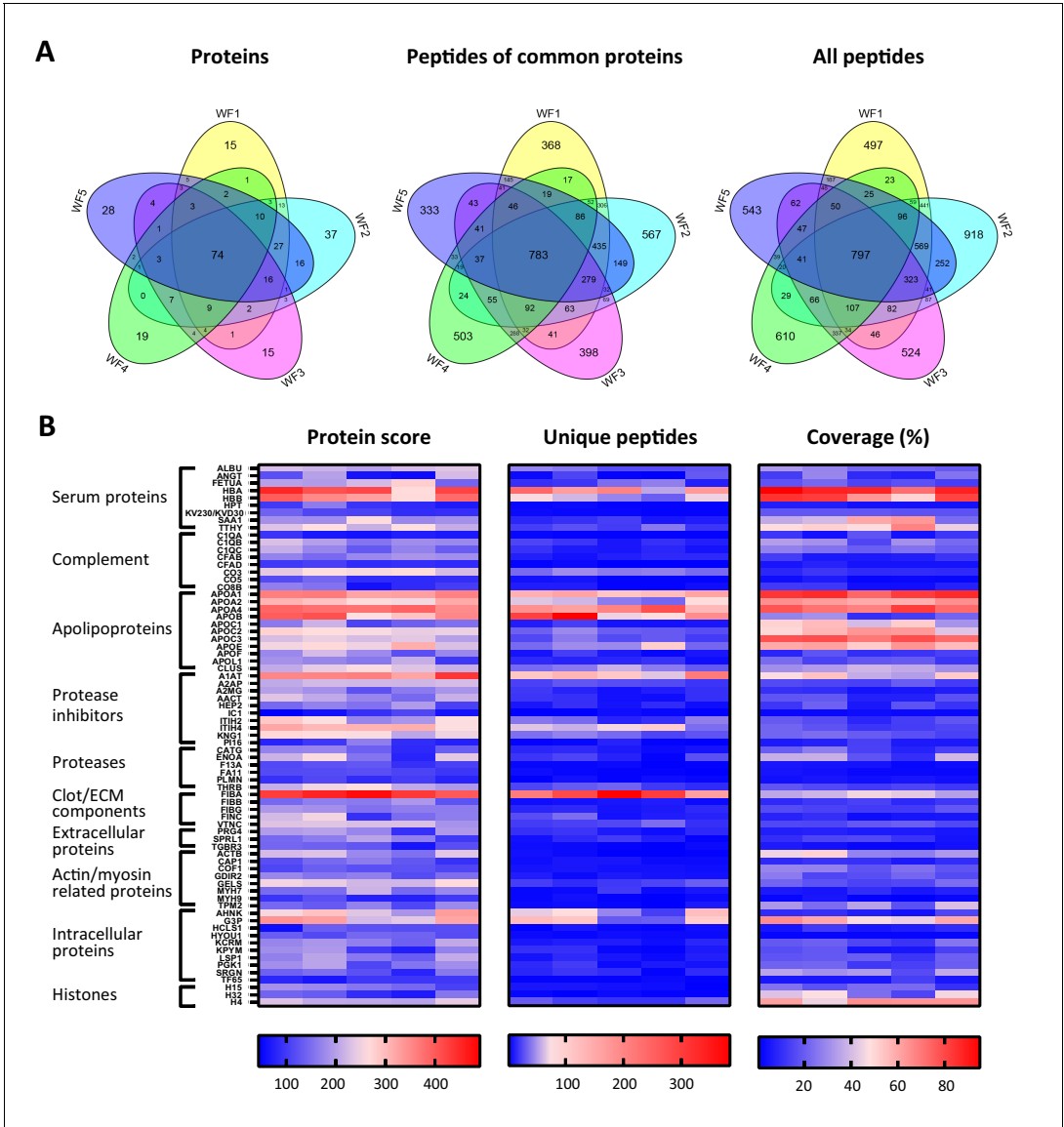

**Figure 4.** Comparison of five acute wound fluids. Peptides were extracted from 100 µL wound fluid in 6 M urea supplemented with 0.05% *Rapi*Gest using the workflow shown in *Figure 1*. (A) Comparison of five wound fluids using Venn diagrams depicting proteins, unique peptides of the 74 proteins common for all five wound fluids and unique peptides of all identified proteins. (B) Heatmaps comparing the five wound fluids depicting the protein score, the number of unique peptides per protein, and the percentage of total coverage of each protein by the identified peptides. Results are shown as combined data of four injections per sample.

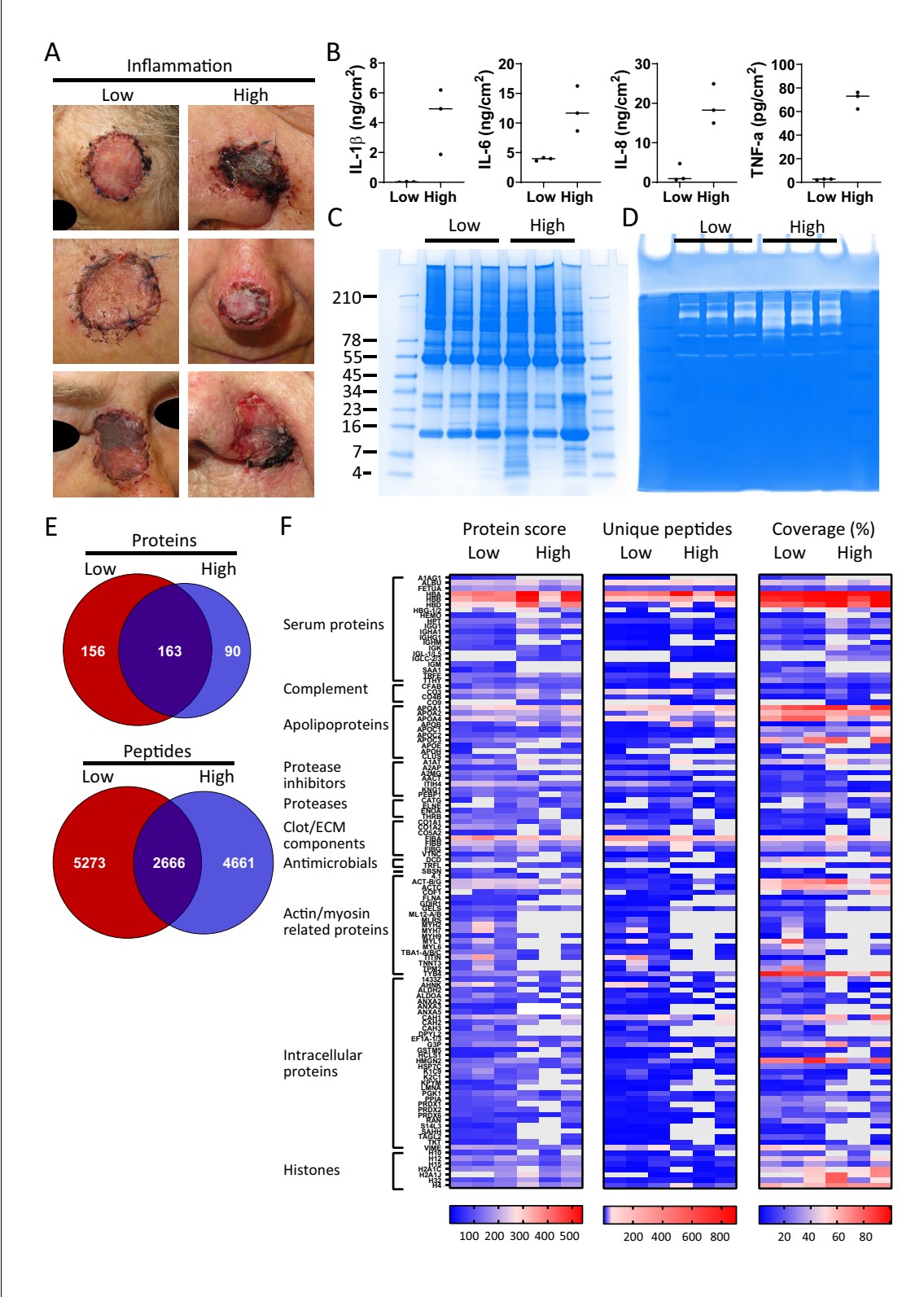

**Figure 5.** Comparison of healing and non-healing infected wounds. (**A**) Photos of wounds, 7 days after surgery, of six patients who had undergone facial full-thickness skin grafting. On the left side are the three that healed well and showed low inflammation and no infection, while on the right side wounds are depicted that were highly inflamed and infected with i.a. *Staphylococcus aureus*. Dressing extracts were made of the seven day old dressings derived from each wound and analysed for cytokine content (**B**), protein and peptide composition using SDS–PAGE (**C**), and enzymatic

*Figure 5 continued on next page*

*Figure 5 continued*

activity using zymograms (D). Peptides were extracted from 280 µg of wound dressing extract in 6M urea supplemented with 0.05% *RapiGest* using the workflow shown in *Figure 1*. (E) Comparison of the pooled results of the three low inflammation samples with the three high inflammation samples using Venn diagrams depicting total number of identified proteins and their unique peptides. (F) Heatmaps comparing the six samples depicting the protein score, the number of unique peptides per protein, and the percentage of total coverage of each protein by the identified peptides. Results are shown as combined data of two independent sample preparations with two injections per sample. Notably, (A) and (B) are derived from previously published results. Figure 5A is reproduced from Figure 2 of *Saleh et al., 2019*, and Figure 5B has been adapted from Figure 4A of *Saleh et al., 2019*. The online version of this article includes the following figure supplement(s) for figure 5:

**Figure supplement 1.** Venn diagrams comparing the numbers of identified proteins and their peptides of each sample in the low and high inflammation groups.

injections per sample (two injections for each of the duplicate extracts) were merged and then subjected to a database search. All identified peptides are listed in Supplementary Dataset 3, while a summary of the total numbers of identified unique peptides, proteins, and the average length and number of amino acids (AA) is given in *Supplementary file 3*. Interestingly, when pooling the results of the three patients per group, we found that 163 proteins (2666 peptides) were found in both groups, although not necessarily in each single patient, whereas 156 proteins (5273 peptides) were only found in the low inflammation group versus 90 (4661 peptides) in the high inflammation group (*Figure 5E*). These results suggest a higher degree of proteolysis in the high inflammation group, as per identified protein an average of 52 unique peptides was found as compared to 34 in the low inflammation group. Indeed, this reflects the expected increase of endogenous proteases in the infected and inflamed wounds due to the influx of immune cells, such as neutrophils and macrophages, combined with the presence of bacterial proteases. Furthermore, the individual samples within the high inflammation group show a higher degree of variability as only 23% of the proteins (10% of all unique peptides) were found in all three patient dressing extracts versus 34% (16% of the peptides) for the low inflammation group (*Figure 5—figure supplement 1*). This variability is further visualised in heatmaps, generated of the proteins that were found in all three extracts of either the low inflammation group or the high inflammation group (*Figure 5F*). As expected, some proteins, such as the serum proteins albumin (ALBU) and hemoglobin (HBA/HBB/HBD), are identified in all wound fluids and dressing extracts (heatmaps, *Figures 2–5*), whereas complement component C9 (CO9), which is a constituent of the membrane attack complex that plays a key role in the innate immune response against bacteria, is only found in fluids from infected wounds. Interestingly, the antimicrobial protein dermcidin (DCD), constitutively expressed by eccrine sweat glands, was detected in the low inflammation group, whereas the protein was absent in the high inflammation group. This may be explained by the reported proteolytic degradation of dermcidin by staphylococci (*Lai et al., 2007*), to such an extent that the remaining peptides can no longer be detected by LC–MS/MS. Contrastingly, the antimicrobial protein lactoferrin (TRFL) was only detected in the high inflammation group. As lactoferrin is secreted by neutrophils upon degranulation, and influx of neutrophils is extensive in inflamed wounds, these results are as expected.

To further investigate protein fragmentation, peptide profiles and peptide alignment maps were generated in order to visualise the qualitative changes at the peptide level. *Figures 6–8* show selected peptide profiles representing proteins of relevance for fibrin formation, wound healing and inflammation, as well as antimicrobial defence. Notably, these profiles do not contain peptides that are post translationally modified. Furthermore, the depicted peptide alignment maps are only showing a selection of the sequences detected for the protein areas that are highlighted by blue boxes.

As expected, peptides derived from the three different fibrinogen chains alpha, beta, and gamma were identified (*Figure 6*). Peptides similar to clinically relevant fibrin degradation products, composed of fibrinopeptides, were found. Furthermore, the antimicrobial and anti-inflammatory peptide GHR28 from the beta chain (*Påhlman et al., 2013*) was identified in the three non-inflamed wounds, whereas only smaller fragments were detected in the inflamed and infected wounds.

Prothrombin is a major protease activated during wound healing, and it was notable that peptides from the thrombin light chain were observed in all three wound samples from non-infected normally healing wounds, whereas they were absent in two of the infected wound fluids. Furthermore, fragments derived from thrombin-derived C-terminal peptides (TCPs), previously found to exert antimicrobial and immunomodulatory effects in wounds (*Saravanan et al., 2017*; *Papareddy et al.,*

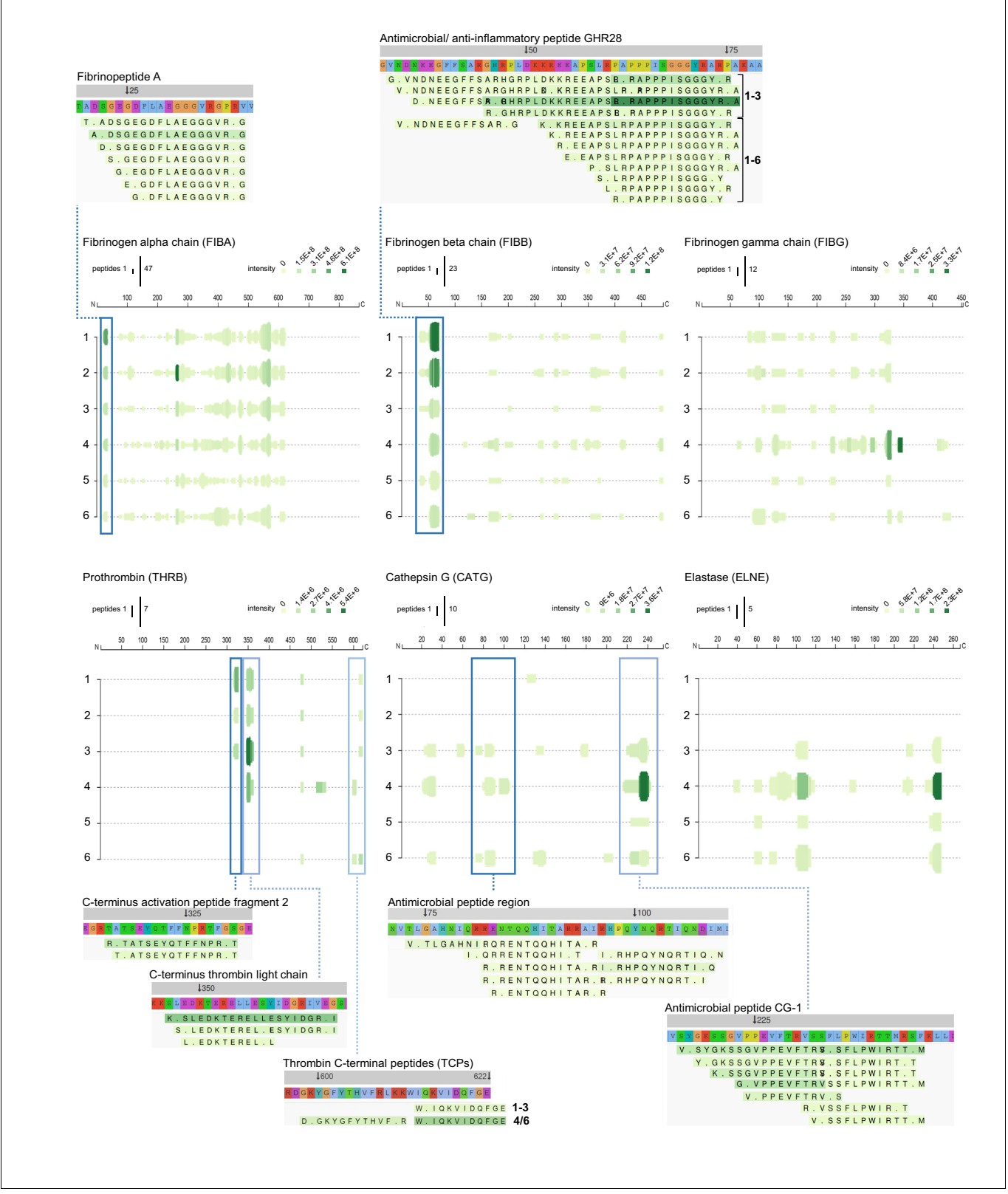

**Figure 6.** Peptide profiles of fibrinogen and selected proteases. Peptide profiles and peptide alignment maps of three fibrinogen chains and the proteases prothrombin, cathepsin G, and neutrophil elastase, were generated from the UniProt IDs, peptide sequences, start and end, and intensities for each protein using the web-based application Peptigram. The height of the green bars is proportional to the number of peptides overlapping the amino acid residue, while the intensity of the colour (green) is proportional to the sum of the intensities overlapping this position. Interesting peptide

*Figure 6 continued on next page*

*Figure 6 continued*

regions are highlighted by blue boxes, and of the corresponding peptides, one sequence of each identified N-terminal is shown for illustration purposes: 1–3, low inflammation samples and 4–6, high inflammation samples.

*2010*; *van der Plas et al., 2016*; *Hansen et al., 2017*), were identified. Whereas thrombin is a major liver-derived enzyme, others such as cathepsin G and elastase are secreted from neutrophils during inflammation and infection. In agreement, peptide fragments from these two enzymes were particularly observed in samples from infected wounds. For cathepsin G, peptides were identified from two antimicrobial regions. Whereas the first region was discovered using synthetic peptides (*Shafer et al., 1993*), the second region corresponds to the previously described antimicrobial peptide CG-1, which was identified in granule extracts from neutrophils (*Miyasaki et al., 1995*). Interestingly, a similar C-terminal fragmentation pattern was identified in neutrophil elastase. Taken together, these results indicate that specific degradation patterns of major plasma and neutrophil proteases can be detected using wound extracts derived from dressings after surgery.

High-molecular-weight kininogen (HMWK) is a multifunctional 120 kDa glycoprotein found in plasma and in granules of platelets (*Schmaier et al., 1983*). The protein is composed of six domains, each having different properties and specific ligands (*Colman and Schmaier, 1997*). Whereas domains 2 and 3 have cysteine protease inhibitor activities, the D4 domain contains the bradykinin sequence, which is released by plasma kallikreins during contact activation and actions of proteases

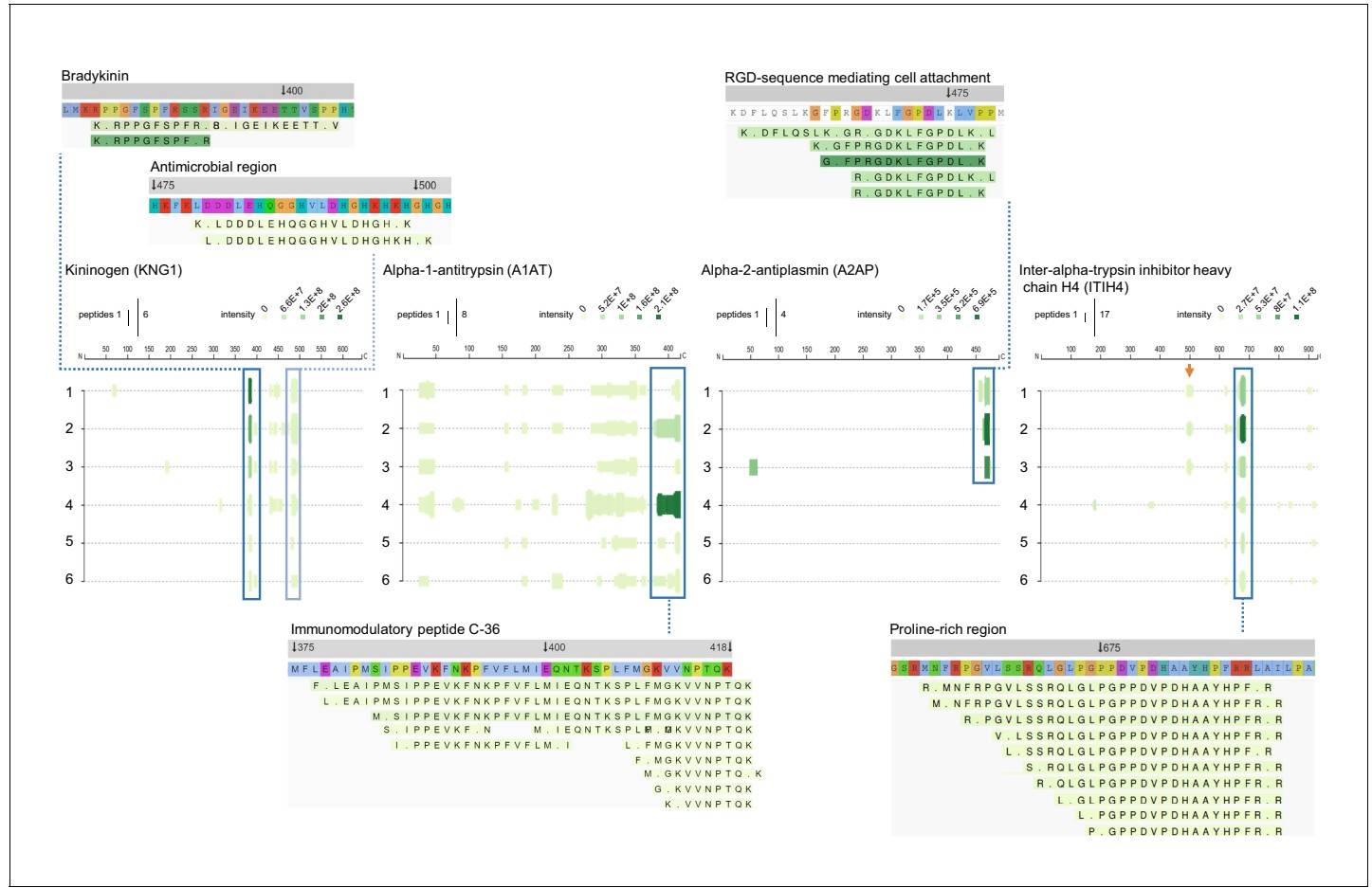

**Figure 7.** Peptide profiles of selected protease inhibitors. Peptide profiles and peptide alignment maps were generated for the protease inhibitors kininogen, alpha-1-antitrypsin, alpha-2-antiplasmin, and inter-alpha-trypsin inhibitor heavy chain H4. Interesting peptide regions are highlighted by blue boxes, and a selection of the corresponding peptides is shown for illustration purposes: 1–3, low inflammation samples and 4–6, high inflammation samples. The orange arrow indicates non-highlighted peptide sequences unique to the three low inflammation samples.

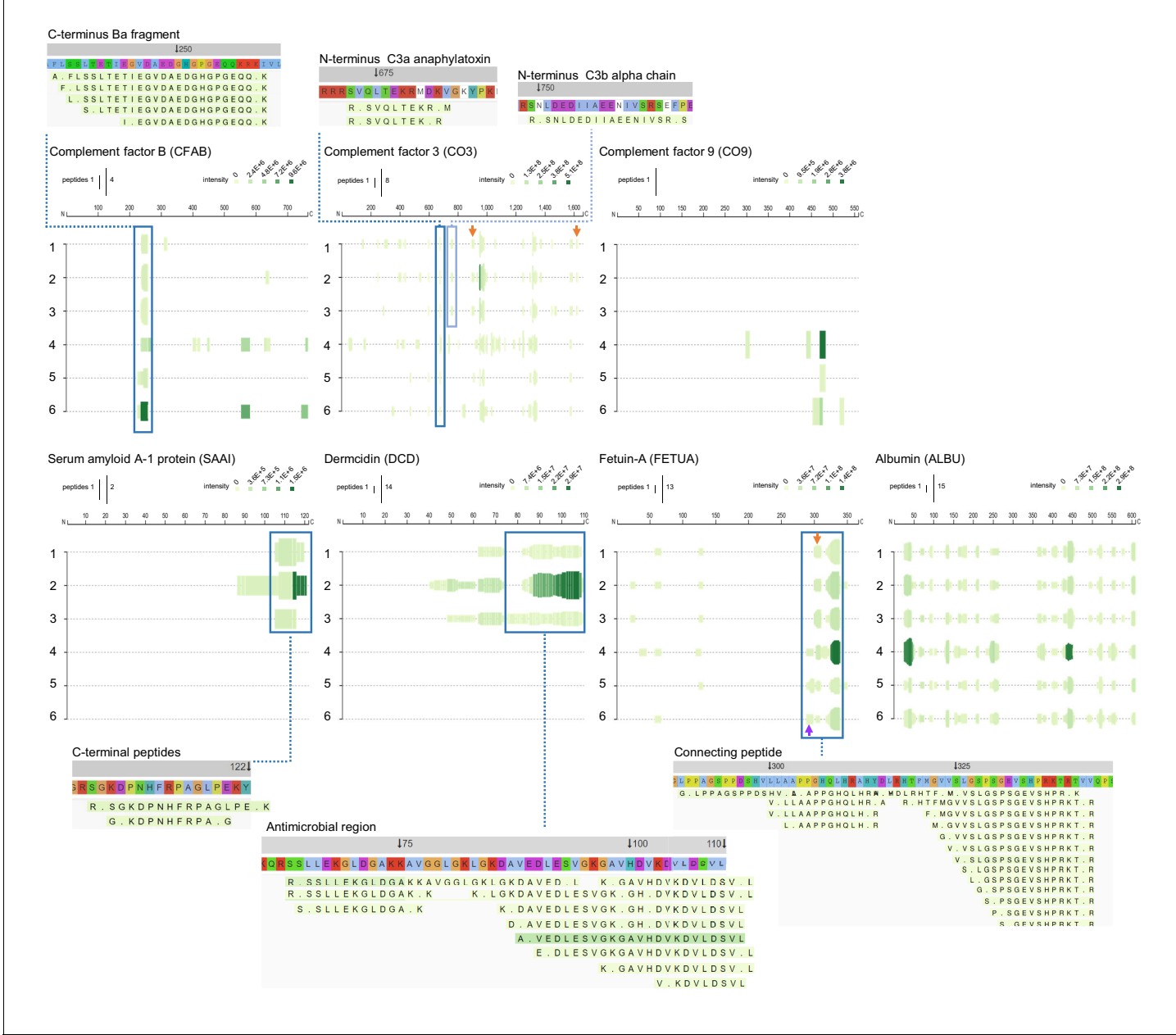

**Figure 8.** Peptide profiles of selected complement factors and additional proteins. Peptide profiles and peptide alignment maps were generated for the complement factors B, C3, and C9, as well as the proteins serum amyloid A-1, dermcidin, fetuin-A, and albumin. Interesting peptide regions are highlighted by blue boxes, and a selection of the corresponding peptides is shown for illustration purposes: 1–3, low inflammation samples and 4–6, high inflammation samples. The orange arrows indicate non-highlighted peptide sequences unique to the three low inflammation samples, while the purple arrow indicates those to the high inflammation group.

such as neutrophil elastase (*Kozik et al., 1998*). Thus, limited proteolysis of HMWK generates highly vasoactive and pro-inflammatory peptides, which are formed at sites of tissue injury and inflammation. The cell-binding D5 from HMWK contains regions dominated by histidine, glycine, and in certain parts, interspersed lysine residues, which also can mediate antimicrobial effects (*Nordahl et al., 2005*). As seen in *Figure 7*, these peptide fragments were indeed observed in the postoperative wounds. Further evidence for the usefulness of the methodology is provided by the detection of fragmentation patterns from additional protease inhibitors such as alpha-1-antitrypsin, antiplasmin, as well as inter-alpha-trypsin inhibitor heavy chain H4, all yielding peptide sequences from distinct

regions of these inhibitors, thereby reflecting the involvement of these inhibitors in direct and specific protease control (*Janciauskiene et al., 2018*; *Lijnen, 2006*; *Lord et al., 2020*), their sensitivity to endogenous and bacterial proteases, and the resulting peptide fragments respective metabolism and turnover in the complex wound environment.

Cleavage of A1AT occurs at the reactive centre loop by proteases such as neutrophil elastase, although MMPs and *S. aureus* enzymes can also cleave the antiproteinase (*Nelson et al., 1998*; *Potempa et al., 1986*). Hence A1AT may be reporting wound-derived protease activity. Moreover, cleavage between Phe$^{376}$-Leu$^{377}$ generates a carboxy-terminal 42-residue peptide, and variants of this fragment, such as a C-terminal 36 aa peptide, have been identified in several human tissues, where they may exert various immunomodulatory functions (*Blaurock et al., 2016*; *Subramaniyam et al., 2006*). Notably, such fragments were also identified in the wound fluids from surgical wounds. Interestingly, C-terminal regions of antiplasmin were only observed in normally healing wounds, and it is notable that this region of plasmin contains bioactive epitopes that can modulate urokinase activity (*Lee et al., 2002*) and interact with endothelial cells (*Thomas et al., 2007*). Recently, C-terminal fragments generated in vivo were indeed identified (*Abdul et al., 2020*). Inter-alpha-trypsin inhibitor heavy chain H4 is a 120 kDa serum glycoprotein secreted primarily by the liver. Peptides derived from the proline-rich region may be biomarkers for a variety of disease states including breast cancer (*van den Broek et al., 2010*), further illustrating that surgical wounds contain regions of biological and diagnostic importance. Notably, peptides from residues 488–504 were only identified in normally healing wounds (*Figure 7*, orange arrow).

In addition to the proteases and protease inhibitors above, the complement cascade is another fundamental defence system activated during wounding and infection. Factor B is part of the alternate pathway of the complement system is cleaved by factor D into two fragments: Ba and Bb. Bb, a serine protease, then combines with complement factor 3b to generate the C3 or C5 convertase. Inspection of the peptide profile for this factor showed that fragments from particularly the S1 peptidase domain were present in the infected wound (*Figure 8*). For the major complement component 3 (CO3), a series of similar fragmentation patterns were identified, with peptides originating in different regions of CO3. Importantly, N-terminals of the well-known anaphylatoxin C3a, which harbours antimicrobial activity (*Nordahl et al., 2004*), as well as the opsonin C3b were identified. Besides common fragments found for all six patients, two peptide sequences were only found in normally healing wounds (orange arrows). As mentioned, complement component 9, which is part of the membrane attack complex that plays a key role in the immune response against bacteria, was only detected in fluids from infected wounds.

Further evidence for distinct proteolytic events affecting selected proteins is seen in the peptigrams representing serum amyloid A-1 (SAA1) peptides, as well as dermcidin (DCD)-derived fragments, which both were only found in non-infected wounds (*Figure 8*, lower panels). SAA1 can upregulate inflammatory cytokines, whereas SAA1 peptides suppress inflammation (*Zhou et al., 2017*). Moreover, the C-terminal SAA (86–104) binds and inhibits human cystatin and carboxy-terminal fragments are involved in amyloid diseases (*Maszota et al., 2015*). Interestingly, similar C-terminal fragments were found in non-infected wounds, indicating that SAA is proteolyzed in wounds, suggesting a yet undisclosed physiological role for such fragments during normal wound healing. DCD is an antimicrobial peptide found in skin (*Schittek et al., 2001*). Proteolytic processing of DCD gives rise to several truncated peptides (*Steffen et al., 2006*). Interestingly, variants of the peptide DCD-1L having the N-terminal SSLLEK (*Rieg et al., 2005*) were indeed identified in normally healing surgical wounds. A proteolytically processed form of fetuin-A, also denoted α2-HS glycoprotein, lacking a segment of 40 amino acid residues bridging its heavy and light chain portions ('connecting peptide') has been described, suggesting that this peptide is released by post-translational processing to fulfil biological role(s) of fetuin-A. The connecting peptide region, which is highly susceptible to proteolytic breakdown in vitro and in vivo (*Nawratil et al., 1996*), was present in the surgical wound fluids. Interestingly, peptides starting with the N-terminal LPPAGS were only detected in the three inflamed wounds (purple arrow), whereas those starting with LLAAPP were unique for the three non-inflamed wounds (orange arrow). Finally, to show the validity and robustness of the method, the peptide profile of serum albumin is presented, showing a high similarity between the fragmentation patterns in the individual wounds.

## Discussion

Endogenous peptides serve as biomarkers of disease progression for several different diseases, and the here presented strategy and methodology identified the wound fluid peptidome as a source for assessment of wound fluid dynamics, as these fragments represent proteolytic events that are the result of basic physiological processes involving coagulation and complement activation, but also additional proteolytic activities by endogenous and bacterial proteases. Degradation of proteins occurs due to the action of set of a proteases and biological processes that results in a low-molecular-weight fraction of endogenous peptides with specific cleavage points. For the analysis of the wound fluid peptidome by mass spectrometry, it is necessary to extract and enrich the peptides, thereby excluding proteins and other interfering molecules. Several different protocols and methods are available for extraction of peptides from biofluids prior to the mass spectrometry measurement. In our work, we have chosen to investigate the peptidome of wound fluids by a combination of low-molecular-weight filtration, solid phase purification, and enrichment followed by mass spectrometry detection and database search. Although solubility of the peptides and the binding of peptides to proteins will affect the outcome of the filtration procedure, previously it was found that solubilisation using urea was efficient for the extraction recovery of endogenous peptides (*Secher et al., 2016*). The LC–MS/MS analysis was performed using data-dependent analysis, where the instrument is set to sequence as many peptides as possible for each sample and the same sample was injected in duplicates. The chosen method allowed us to characterise more than 7800 peptides in acute wound fluids derived from the degradation of over 370 proteins. We have focused our subsequent analysis on a subset of peptides corresponding from the degradation of the most abundant proteins that could be measured in all of the examined acute wound fluids. The definition of the wound peptidome of acute wounds in well-defined sterile wound fluids validated the approach, identifying multiple peptides derived from several protein families. The following analyses on dressing extracts from skin-grafted surgical wounds provided a proof-of-principle, showing the possibility of analysis of selected peptide regions as potential biomarkers for inflammation and wound healing. This is of high relevance from a clinical perspective, as skin grafting surgery is normally associated with a higher risk of SSI (*Saleh and Schmidtchen, 2015*; *Saleh et al., 2019*), motivating the use of such wound fluids from this type of dermatologic surgery.

As briefly mentioned in the introduction, classical proteomics approaches have been employed in the study of different wound types. *Eming et al., 2010* studied the distribution of tissue repair proteins in exudates of healing acute and non-healing venous wounds on the legs. Subsequent studies have added to the categorisation of proteins identified in various wounds such as diabetic and pressure ulcers (*Broadbent et al., 2010*; *Krisp et al., 2013*; *Wyffels et al., 2010*; *Steinsträßer et al., 2010*; *Edsberg et al., 2012*). Auf dem Keller and colleagues applied quantitative proteomics strategies to dissect proteolytic pathways during wounding and the activity of distinct protease groups along the healing process. Moreover, they also mapped proteolytic pathways in vivo and established protease–substrate relationships that will help to better understand protease action in cutaneous wound repair and other inflammatory conditions in skin (*Sabino et al., 2018*; *Sabino et al., 2015*; *Sabino, 2017*; *Savickas et al., 2020*). However, it is of note that methods for detection of downstream products reflecting inflammation are rare. Although Auf dem Keller's work involves studies on such protease patterns, their method focuses on 'N-terminomics', while low-molecular-weight peptides, such as the ones studied here, are mainly lost during the preparation steps. Interestingly, at the proteomic level, in their studies on healing and non-healing wounds on the legs, *Eming et al., 2010* showed that the serine protease thrombin, as well as the antimicrobial DCD, was exclusively detected in healing wounds. In contrast, the neutrophil-derived lactoferrin was increased in non-healing wounds. Moreover, the three major fibrinogen chains alpha, beta, and gamma were dominating in the wound fluids from all wound types. Although the wounds types are obviously different with respect to localisation and pathogenesis, the clinical grading into non-healing and healing is the same for both studies. It is therefore notable that corresponding observations on the above proteins were indeed made in this study at the peptidome level as well, lending further support for the diagnostic potential of the here described methodology. The significance of the methodology is further underscored by recent bioinformatics applications based on the present datasets (*Hartman et al., 2021*). Thus, in a global approach, algorithms in Python as well as open source softwares such DeepAmPEP30 (*Yan et al., 2020*) and Proteasix (*Klein et al., 2013*) were employed. Analyses of

distribution of amino acids at N- and C-terminal positions, the corresponding four amino acid long terminal segments, and in the complete sequence were conducted. Wound fluids derived from dressings were found to contain larger proportion of acidic residues in the P1 position. Moreover, specific sequences like HKYH, LERM, and SKYR were especially prevalent at the C-terminal in infected wounds. In silico prediction using Proteasix showed that cleavages by proteases such as MMP 7, MMP 9, and neutrophil elastase were most prevalent, and in particular neutrophil elastase was particularly up regulated in infected wound samples (*Hartman et al., 2021*), serving as a possible indicator of wound infection. DeepAMP identified novel antimicrobial peptides in hemoglobin, and moreover, peptides derived from LPS-binding regions of hemoglobin were identified in infected surgical wounds. Apart from demonstrating the power of data-driven approaches to wound fluid peptidomics, the results also highlight the usefulness of novel bioinformatic tools for analysis of the vast datasets obtained using the methodology disclosed in this report.

Today, the current method for characterisation of infection and related wound complications in the clinic is first to assess wound status according to clinical experience combined with bacterial detection. There are many obvious signs of advanced infection including redness, heat, swelling, purulent exudate, smell, and pain, but the challenge is to translate these observations to objective and sensitive methods that correctly evaluate wound status and, in particular, the associated inflammation, before wounds even reach this dysfunctional state. Moreover, the presence of bacteria in wounds does not correlate with wound inflammation and infection. For example, non-healing ulcers are frequently colonised by staphylococci, even without infection, and post-surgical wounds can also harbour *S. aureus* without signs of clinical infection (*Saleh et al., 2019*; *Grice and Segre, 2012*; *Johnson et al., 2018*). Therefore, given that mere bacterial presence is not discriminatory for wound infection, there is a clear need for objective methods that enable detection of dysfunctional wound healing to better guide the use of treatment. Approaches under development today to evaluate bacteria and the concomitant inflammation are use of biological or chemical sensors of wound exudates to detect bacterial antigens, monitor pH, temperature, oxygen, and enzymes. Spectroscopic and imaging techniques are also possible as future advanced wound monitoring techniques (*Dargaville et al., 2013*; *World Union of Wound Healing Societies (WUWHS), 2008*; *Li et al., 2021*; *Lindley et al., 2016*). Methods that measure host responses as one way to assess wound healing, and particularly inflammation, have also been developed. For example, the major enzymes from neutrophils, human neutrophil elastase (HNE), and cathepsin G have been reported as early-stage warning markers for non-healing ulcers. In the clinic, detection of HNE or MMPs is used by a device developed by Systagenix, measuring elevated protease activity through their Woundcheck Protease Status diagnostic. In a recent study, it was identified that wounds that had a clinical appearance of being more inflamed indeed had higher levels of proteases, as demonstrated for the major MMPs, MMP-2 and −9. These wounds had increased cytokine levels relative to the wound surface area, particularly observed for IL-1β, but also for TNF-α (*Saleh et al., 2019*). Moreover, these wounds also had significantly higher overall bacterial loads, a factor of importance in the development of SSIs. As demonstrated in this report, apart from the cytokines IL-1β, IL-6, IL-8, and TNF-α, a vast number of potential new peptide sequences could be detected in these wounds, using a qualitative approach focusing on major protein families involved in coagulation, complement activation, and protease control. Furthermore, subsequent mining of the peptidome data sets with bioinformatics methods identified additional subtle sequence differences (*Hartman et al., 2021*). Together, these results provide a proof-of-principle for utilisation of wound fluid peptidomics in future biomarker-oriented studies on larger well-defined patient groups. Combining the use of such new peptides as biomarkers along with classical analyses of cytokines and MMPs could yield higher diagnostic sensitivity and specificity with respect to wound status and infection risk. Moreover, the development of sensors, based on antibodies or aptamers that target selected peptides, could make this diagnostic approach more attractive in the clinics, enabling fast readouts and thus facilitate the needed clinical studies. Finally, the peptidomics data generated here provides previously undisclosed data on proteolytic fragmentation patterns during wounding, which may aid in the discovery of novel bioactive peptides and elucidation of their biological roles during wound healing.

## Materials and methods

### Study approval

This study was carried out in accordance with the recommendations of the Ethics Committee at Lund University, Lund, Sweden, with written informed consent from all subjects in accordance with the Declaration of Helsinki. The protocols for the use of human blood (permit no. 657–2008) and human wound materials (708–01, 509–01, and 762–2013) were approved by the Ethics Committee at Lund University.

### Sample collection

Plasma was collected from citrated venous blood from three healthy donors by centrifugation at 2000 g. Sterile acute wound fluids, obtained from surgical drainages after mastectomy from five donors, were collected for 24 hr, 24–48 hr after surgery followed by centrifugation as described previously (*Lundqvist et al., 2004*). Wound fluids from six patients that underwent facial full-thickness skin grafting were extracted from Mepilex (Mölnlycke Health Care, Sweden) tie-over wound dressings, which had been on the wound for 1 week, as described (*Saleh et al., 2019*). In short, wound exudate was extracted from dressings after incubation in 10 mM Tris (pH 7.4) for 1 hr at 8°C while shaking. All samples were stored at −20°C before use.

### Peptide extraction

Frozen plasma and/or sterile acute wound fluids were defrosted and mixed with three parts freshly made 8 M urea (in 10 mM Tris, pH 7.4, yielding a final urea concentration of 6 M) alone or supplemented with *Rapi*Gest SF (0.05% final concentration; Waters, USA) or trifluoroacetic acid (0.1% final concentration; Sigma-Aldrich, Germany) and incubated for 30 min at room temperature (RT) followed by size exclusion. For this purpose, centrifugal filters with 30 kDa cut-off (Microcon 30, regenerated cellulose, Millipore, Ireland) were first rinsed with 100 µL buffer and centrifuged at 14,000 g for 15 min at RT, then loaded with the incubated sample mixtures, followed by 30 min centrifugation (14,000 g at RT) and a final filter washing step with 100 µL of extraction buffer (5 min, 14,000 g). Due to large variations in protein content of the dressing extracts (between 2.8 and 16.5 mg/mL), which was measured using the Pierce BCA Protein Assay Kit (Thermo Scientific, USA) according to manufacturer's instructions, protein concentrations were standardised. For this purpose, 10 mM Tris was added to 280 µg of each fluid to obtain a sample volume of 100 µL in total, which was then incubated with 300 µL 8 M urea supplemented with *Rapi*Gest SF for 30 min, followed by the size exclusion steps as described above, with the modification that the filters were centrifuged at 10,000 g. For all samples, the two filtrates containing the peptides were pooled and analysed by SDS–gel electrophoresis directly or stored at −20°C before analysis by LC–MS/MS.

### SDS–gel electrophoresis

Extracted peptide samples or 20 µg of each dressing extract was denatured at 85°C for 5 min in 1× SDS sample buffer followed by separation on 10–20% Tris–Tricine mini gels in 1× Tris–Tricine SDS running buffer for 90 min at 125 V. Gels and buffers were derived from Invitrogen (USA). Gels were stained using GelCode Blue Safe Protein Stain (Thermo Scientific, USA) or the SilverQuest Silver Staining Kit (Invitrogen) according to manufacturer's instructions, and patterns were visualised using a Gel Doc Imager (Bio-Rad Laboratories, USA).

### Zymograms

Gels were prepared consisting of a separation gel (0.1% [w/v] gelatine, 0.1% [w/v] SDS, 10% acrylamide in 375 mM Tris buffer [pH 8.8], 0.05% [v/v] TEMED, and 0.05% [w/v] APS) and a stacking gel (0.1% [w/v] SDS, 4% acrylamide in 125 mM buffer [pH 6.8], 0.1% [v/v] TEMED, and 0.05% [w/v] APS). Dressing extracts (5 µg) were mixed with sample buffer (20% [v/v] glycerol, 5% [w/v] SDS, 0.03% [w/v] bromophenol blue, 0.4 M Tris–HCl pH 6.8) in a 1:1 ratio, transferred to the slots, and gels were run in electrophoresis buffer (25 mM Tris, 0.2 M glycine, and 0.5% [w/v] SDS in $H_2O$ at pH 8.7) for 60 min at 150 V. Next, gels were washed in $H_2O$, incubated for 1 hr in 2.5% Triton X-100, washed again, and placed in enzyme buffer (5 mM $CaCl_2$, 1 µM $ZnCl_2$, 200 mM NaCl, and 50 mM Tris–HCl pH 7.5). After overnight incubation at 37°C while shaking (50 rpm), gels were washed and stained using

Coomassie brilliant Blue. Enzymatic activity was observed after destaining the gels with a solution consisting of 10% EtOH and 14% acetic acid in $H_2O$.

## Measurement of cytokine levels

Cytokine levels in the dressing extracts were determined using ELISA kits (R and D systems, USA) according to manufacturer's instructions. Results, adjusted to the total protein concentrations of the extracts, are mean values of triplicate measurements.

## LC–MS/MS analysis

Prior to LC–MS/MS analyses, 80µL of defrosted peptide extracts were acidified with 5 µL of 10% formic acid and then trapped and enriched on StageTip columns (*Rappsilber et al., 2007*). Next, peptides were extracted with 70% ACN and 0.1% TFA and dried down before reconstitution in 20 µL of 2% ACN and 0.1% TFA. For each run, 2 µL of reconstituted sample was injected, which corresponded to 1.6 µL of the original wound fluid or plasma sample. For the evaluation of the sample preparation method, LC–MS/MS analyses were carried out on an Orbitrap Fusion Tribrid MS (Thermo Scientific) as described earlier (*Mehmeti et al., 2019*), with the following modifications. During the elution steps, the percentage of solvent B increased from 5% to 22% in the first 20 min, then increased to 20% in 85 min, then to 30% in 20 min, and to 90% in a further 5 min, where it was kept for 5 min.

Subsequent analysis of patient samples was performed using an HFX Orbitrap MS system (Thermo Scientific) equipped with a Dionnex 3000 Ultimate HPLC (Thermo Fisher, USA). Injected peptides were trapped on an Acclaim PepMap C18 column (3 µm particle size, 75 µm inner diameter $\times$ 20 mm length). After trapping, gradient elution of peptides was performed on an Acclaim PepMap C18 column (100 Å 2 µm, 150 mm, 75 µm). The outlet of the analytical column was coupled directly with the mass spectrometer using a Nano Easy source. The mobile phases for LC separation were 0.1% (v/v) formic acid in LC–MS grade water (solvent A) and 0.1% (v/v) formic acid in acetonitrile (solvent B). Peptides were first loaded onto the trapping column and then eluted to the analytical column with a gradient. The percentage of solvent B increased from 2% to 27% in the first 107 min, then increased to 32% in 10 min, then to 90% in 15 min where it was kept for 5 min. The peptides were introduced into the mass spectrometer via a Stainless steel emitter 40 mm (Thermo Fisher), and a spray voltage of 1.9 kV was applied. The capillary temperature was set at 275°C. Data acquisition was carried out using a top 20 based data-dependent method. MS was conducted in the range of 350–1350 *m/z* at a resolution of 120,000 FWHM. The filling time was set at a maximum of 100 ms with limitation of $3 \times 10^6$ ions. MSMS was acquired with a filling time maximum 300 ms with limitation of $5 \times 10^4$ ions, a precursor ion isolation width of 2.0 *m/z*, and resolution of 15,000 FWHM. Normalised collision energy was set to 28%. Only multiply charged (2+ to 5+) precursor ions were selected for MS2. The dynamic exclusion list was set to 30 s.

## Data analysis

MS/MS spectra were searched with PEAKS software. UniProt Human, including 20,413 protein sequences, was used with nonspecific cleavage; 5 ppm precursor tolerance and 0.5 Da fragment tolerance were used for the experiments conducted with the Fusion instrument and 0.02 Da for the fragments when the HFX was used. Oxidation (M) and deamidation (NQ) were treated as dynamic modification. Search results were filtered by using 1% FDR for the wound fluids, or a 23 score for the plasma samples, and at least two unique peptides for each protein. The mass spectrometry peptidomics data have been deposited to the ProteomeXchange Consortium via the PRIDE (*Perez-Riverol et al., 2019*) partner repository with the dataset identifiers PXD023884 and PXD023244.

For data visualisation, graphs and heatmaps were generated using Graphpad Prism, Venn diagrams were made in VennDis (*Ignatchenko et al., 2015*), while peptide profiles and peptide alignment maps were made using the web-based application Peptigram (*Manguy et al., 2017*).

## Acknowledgements

This work was supported by grants from Alfred Österlunds Foundation, Edvard Welanders Stiftelse and Finsenstiftelsen (Hudfonden), the Knut and Alice Wallenberg Foundation (2011.0037), Lars Hiertas Memorial Foundation, LEO Foundation (project LF18020), O.E. and Edla Johanssons Foundation,

the Royal Physiographic Society in Lund, the Swedish Research Council (projects 2017–02341 and 2020-02016), the Swedish Government Funds for Clinical Research (ALF), the Swedish Strategic Research Foundation, Vinnova and Åke Wibergs Foundation. The funders had no role in study design, data collection and interpretation, or the decision to submit the work for publication.

## Additional information

### Funding

| Funder | Grant reference number | Author |
| --- | --- | --- |
| Alfred Österlunds Stiftelse | | Mariena JA van der Plas<br>Artur Schmidtchen |
| Hudfonden (Edvard Welanders Stiftelse and Finsenstiftelsen) | | Mariena JA van der Plas<br>Artur Schmidtchen |
| Stiftelsen Lars Hiertas Minne | | Mariena JA van der Plas |
| LEO Fondet | LF18020 | Mariena JA van der Plas |
| OE och Edla Johanssons Vetenskapliga Stiftelse | | Mariena JA van der Plas |
| Royal Physiographic Society in Lund | | Mariena JA van der Plas |
| Swedish Research Council | 2017-02341 | Artur Schmidtchen |
| Swedish Strategic Research Foundation | | Artur Schmidtchen |
| Swedish Government Funds for Clinical Research (ALF) | | Artur Schmidtchen |
| Åke Wiberg Stiftelse | | Mariena JA van der Plas |
| Knut and Alice Wallenberg Foundation | 2011.0037 | Artur Schmidtchen |
| VINNOVA | | Artur Schmidtchen |
| Swedish Research Council | 2020-02016 | Artur Schmidtchen |

The funders had no role in study design, data collection and interpretation, or the decision to submit the work for publication.

### Author contributions

Mariena JA van der Plas, Conceptualization, Resources, Data curation, Formal analysis, Supervision, Funding acquisition, Validation, Investigation, Visualization, Methodology, Writing - original draft, Project administration; Jun Cai, Formal analysis, Investigation, Visualization, Writing - review and editing; Jitka Petrlova, Conceptualization, Investigation, Methodology, Writing - review and editing; Karim Saleh, Resources, Investigation, Writing - review and editing; Sven Kjellström, Conceptualization, Resources, Data curation, Formal analysis, Validation, Investigation, Visualization, Methodology, Writing - review and editing; Artur Schmidtchen, Conceptualization, Resources, Supervision, Funding acquisition, Methodology, Writing - original draft, Project administration

### Author ORCIDs

Mariena JA van der Plas  https://orcid.org/0000-0002-3233-2881
Artur Schmidtchen  http://orcid.org/0000-0001-9209-3141

### Ethics

Human subjects: This study was carried out in accordance with the recommendations of the Ethics Committee at Lund University, Lund, Sweden with written informed consent from all subjects in accordance with the Declaration of Helsinki. The protocols for the use of human blood (permit no. 657-2008) and human wound materials (708-01, 509-01 and 762-2013) were approved by the Ethics Committee at Lund University.

Decision letter and Author response
Decision letter https://doi.org/10.7554/eLife.66876.sa1
Author response https://doi.org/10.7554/eLife.66876.sa2

## Additional files

### Supplementary files
• Source data 1. All identified peptides in three plasma samples. List of 770 unique peptide sequences identified by LC–MS/MS.

• Source data 2. All identified peptides in five acute wound fluids. List of 7809 unique peptide sequences identified by LC–MS/MS.

• Source data 3. All identified peptides in six dressing extracts. List of 10,789 unique peptide sequences identified by LC–MS/MS.

• Supplementary file 1. Summary of the mass spectrometry results for plasma, acute wound fluids, and dressing extracts.

• Supplementary file 2. Identified peptides and average length from five acute wound fluids.

• Supplementary file 3. Identified peptides and average length from six dressing extracts.

• Transparent reporting form

### Data availability
The mass spectrometry peptidomics data have been deposited to the ProteomeXchange Consortium (http://proteomecentral.proteomexchange.org) via the PRIDE partner repository with the dataset identifiers PXD023884 and PXD023244.

The following dataset was generated:

| Author(s) | Year | Dataset title | Dataset URL | Database and Identifier |
|---|---|---|---|---|
| van der Plas MJA, Cai J, Petrlova J, Saleh K, Kjellstrom S, Schmidtchen A | 2021 | Method development and characterization of the low molecular weight peptidome of human wound fluids | https://www.ebi.ac.uk/pride/archive/projects/PXD023884 | PRIDE, PXD023884 |

The following previously published dataset was used:

| Author(s) | Year | Dataset title | Dataset URL | Database and Identifier |
|---|---|---|---|---|
| Hartman E, Wallblom K, van der Plas MJA, Petrlova J, Cai J, Saleh K, Kjellstrom S, Schmidtchen A | 2021 | Bioinformatic analysis of the wound peptidome reveals potential biomarkers and antimicrobial peptides | https://www.ebi.ac.uk/pride/archive/projects/PXD023244 | PRIDE, PXD023244 |

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
