## [Decision Letter]

**Acceptance summary:**

This well-done study establishes a work flow for the analysis of the peptidome of wound fluids. By doing so it enables the identification of peptide patterns associated with wounds that are healing versus non-healing. The method may therefore help to define candidate biomarkers for wound healing.

**Decision letter after peer review:**

Thank you for submitting your article "Method development and characterization of the low molecular weight peptidome of human wound fluids" for consideration by *eLife*. Your article has been reviewed by 3 peer reviewers, one of whom is a member of our Board of Reviewing Editors, and the evaluation has been overseen by Mone Zaidi as the Senior Editor. The following individual involved in review of your submission has agreed to reveal their identity: Bouke Boukema (Reviewer #2).

Essential revisions:

1) The main issue that came up during review was the publication from your group of a very similar article in Frontiers in Immunology, which deals with an overlapping topic. There was a significant lack of clarity regarding the novelty of the submitted manuscript versus the recently published article. This issue should be fully addressed in any resubmission.

*Reviewer #1 (Recommendations for the authors):*

The major issue that the authors need to address is discerning differences in this manuscript compared to their recent publication in Frontiers in Immunology, "Bioinformatic Analysis of the Wound Peptidome Reveals Potential Biomarkers and Antimicrobial Peptides".

*Reviewer #3 (Recommendations for the authors):*

From a technological point of view the study is excellent and provides a better understanding on protein degradation and formation of (bioactive) peptides in wound fluids (healing or not).

My enthusiasm is, however, a bit dampened.

First, authors need to discuss the findings of their present study with the study of the same group recently published in Frontiers in Immunology, which deals with a similar and overlapping topic.

Moreover, although I appreciate the work that has been done to establish peptidome characterization of wound fluids and to show differential peptide patterns in healing and non-healing wound exudates, evidence that the technology can be adapted as diagnostic approach for proper wound healing or not is missing. And this makes the manuscript rather technical and perhaps more suited for a more specialized journal. Perhaps this could be addressed also by working out the novelty of the present study in more detail? Is it "only" methods development or what can we learn from the study, what is the real novelty of the findings?

---

## [Author Response]

Essential revisions:1) The main issue that came up during review was the publication from your group of a very similar article in Frontiers in Immunology, which deals with an overlapping topic. There was a significant lack of clarity regarding the novelty of the submitted manuscript versus the recently published article. This issue should be fully addressed in any resubmission.

We appreciate the comments and apologise for not adding the needed clarifications and information on the relationship between this and the subsequent Frontiers Immunology paper. We want to stress that the current *eLife* manuscript under revision here was originally uploaded on the preprint server medRxiv on the 3rd of November 2020. The Frontiers paper, which is a follow up study of the current manuscript, was published in February 2021.

Importantly, the latter is based on and refers to the original methodology and peptidome data described in the medRxiv article (which was then later transferred to *eLife*). For clarity, the text in the Frontiers paper, clearly referring to the medRxiv paper (reference 9), is found below.

In the current revised version of the manuscript, we now clearly describe the originality of the methodology described here and its precedence, and the overall separate and independent character of the current manuscript under review. In particular, we now thoroughly discuss the findings of this present study in relation to the Frontiers article and therefore, we believe that the uniqueness of the present paper is now made very clear. In our opinion, the Frontiers article does not diminish the novelty and strength of the current manuscript. Instead, it increases its strength as we showed that the here described method and obtained qualitative results can be used successfully in quantitative bioinformatics analysis as well.

Section two in the introduction of the Frontiers Immunology paper:

“We have recently developed a peptidomics method for the characterization of endogenous peptides of wound fluids. […] As antimicrobial defense and innate immunity is intimately linked to wound healing another goal of the work was to explore whether there could be alterations in global patterns of antimicrobial peptides (AMP).”

Reviewer #1 (Recommendations for the authors):The major issue that the authors need to address is discerning differences in this manuscript compared to their recent publication in Frontiers in Immunology, "Bioinformatic Analysis of the Wound Peptidome Reveals Potential Biomarkers and Antimicrobial Peptides".

We appreciate the comments and apologise for not adding the needed clarifications and information on the relationship between this and the subsequent Frontiers Immunology paper. We want to stress that the current *eLife* manuscript under revision here was originally uploaded on the preprint server medRxiv on the 3rd of November 2020. The Frontiers paper, which is a follow up study of the current manuscript, was published in February 2021.

Importantly, the latter is based on and refers to the original methodology and peptidome data described in the medRxiv article (which was then later transferred to *eLife*). The Frontiers article utilises part of the in this study obtained peptidome data sets, which were uploaded on the ProteomeXchange server. We have added significant revisions in the current manuscript, so the relationship between the studies is now clarified (lines 105-108 and lines 393-409).

Thus, while in the current manuscript the focus is on method development and qualitative differences in identified proteins and peptides, in the separate Frontiers Immunology paper, the uploaded data sets derived from the present report were further mined and processed using global bioinformatic approaches exploring quantitative differences in the identified peptidomes, demonstrating the applicability of the peptidome data generated in this study. Indeed, the significance of the methodology presented in the *eLife* submission is further underscored by the subsequent bioinformatics applications in the Frontiers paper (algorithms in Python as well as open source softwares such Deep-AmPEP30 and Proteasix), based on the datasets described in the current manuscript. Apart from demonstrating the power of data driven approaches to wound fluid peptidomes, the results also highlight the usefulness of novel bioinformatic tools for analysis of the vast datasets obtained using the methodology disclosed in the present *eLife* manuscript under revision. Taken together, we believe that the methodology described here and its precedence, and the overall separate and independent character of the article under review in *eLife* is now clarified.

Reviewer #3 (Recommendations for the authors):From a technological point of view the study is excellent and provides a better understanding on protein degradation and formation of (bioactive) peptides in wound fluids (healing or not).My enthusiasm is, however, a bit dampened.First, authors need to discuss the findings of their present study with the study of the same group recently published in Frontiers in Immunology, which deals with a similar and overlapping topic.

We appreciate the comments and apologise for not adding the needed clarifications and information on the relationship between this and the subsequent Frontiers Immunology paper. We want to stress that the current *eLife* manuscript under revision here was originally uploaded on the preprint server medRxiv on the 3rd of November 2020. The Frontiers paper, which is a follow up study of the current manuscript, was published in February 2021.

Importantly, the latter is based on and refers to the original methodology and peptidome data described in the medRxiv article (which was then later transferred to *eLife*). The Frontiers article utilises part of the in this study obtained peptidome data sets, which were uploaded on the ProteomeXchange server. We have added significant revisions in the current manuscript, so the relationship between the studies is now clarified (lines 105-108 and 393-409).

Thus, while in the current manuscript the focus is on method development and *qualitative* differences in identified proteins and peptides, in the separate Frontiers Immunology paper, the uploaded data sets derived from the present report were further mined and processed using global bioinformatic approaches exploring *quantitative* differences in the identified peptidomes, demonstrating the applicability of the peptidome data generated in this study.

Indeed, the significance of the methodology presented in the *eLife* submission is further underscored by the subsequent bioinformatics applications in the Frontiers paper (algorithms in Python as well as open source softwares such Deep-AmPEP30 and Proteasix), based on the datasets described in the current manuscript. Apart from demonstrating the power of data driven approaches to wound fluid peptidomes, the results also highlight the usefulness of novel bioinformatic tools for analysis of the vast datasets obtained using the methodology disclosed in the present *eLife* manuscript under revision. Taken together, we believe that the methodology described here and its precedence, and the overall separate and independent character of the article under review in *eLife* is now clarified.

Moreover, although I appreciate the work that has been done to establish peptidome characterization of wound fluids and to show differential peptide patterns in healing and non-healing wound exudates, evidence that the technology can be adapted as diagnostic approach for proper wound healing or not is missing. And this makes the manuscript rather technical and perhaps more suited for a more specialized journal. Perhaps this could be addressed also by working out the novelty of the present study in more detail? Is it "only" methods development or what can we learn from the study, what is the real novelty of the findings?

We appreciate the comments of the reviewer which show we haven’t been clear enough in our discussion. The goal of our study and method is not to adapt the technology itself as a diagnostic approach, as that would be too unpractical and expensive for frequent use in a broad clinical setting, but instead to use the technology to discover novel diagnostic/ prognostic biomarkers that subsequently can be developed into easy to use lab-on-a-chip based technologies. We added this to the discussion (lines 445-448). In this respect, many endogenous peptides have been reported as potential biomarkers of various diseases. An example is the HBB peptide AGVANALAHKYH, which is a urinary biomarker for psoriatic arthritis, but not for other inflammatory arthritis subgroups (Siebert et al., Sci Rep 2017; 7:40473), We found this specific sequence in our samples; almost three times more often identified in the high inflammation group than in the low inflammation group.

Regarding the novelty of the study, although essential, method development is only part of it of course. From a biological and clinical perspective, the real interesting data is shown in figures 5-8. We show clear differences between the low and high inflammation groups, with various proteins and peptide sequences that were exclusively identified in the samples of one of these groups. For example, we found that the antimicrobial peptide dermcidin (DCD) was detected in the dressing extracts of all samples from the low inflammation group, whereas it was absent in the high inflammation samples. In a previous report by Eming et al. (J Proteome Res. 2010;9(9):4758-66), it was shown at the proteomic level that DCD was exclusively detected in healing wounds (lines 384-386). Therefore, this protein is a promising biomarker for healing. Whereas this is an example of a protein that corroborates findings in a previous report, we identified various other proteins and peptide sequences that are unique to one patient group, many which have not been described previously, which all are potential biomarkers. Especially these unique peptides are of interest, as with our method we can distinguish specific peptide sequences between groups, even when a protein is present in all samples. Furthermore, our results may help explaining some of the processes during wound healing and infection. An obvious example of this is the identification of CATG, ELNE and TRFL in the ‘high inflammation’ dressing extracts, whereas these proteins were mostly absent in the low inflammation group. As these proteins are all released by neutrophils, this is a clear indicator that there is inflammation in these wounds. Notably, they have been reported as early stage warning markers for non-healing ulcers. Another example is the identification of complement factor C9 in the samples of the high inflammation group, which could be a marker for infection, as it is only released in the presence of bacteria.

Taken together, although we cannot make strong conclusions due to the small sample size, that doesn’t mean that the only novelty of our manuscript is method development. By contrast, we have identified many novel peptide sequences, which provide previously undisclosed proteolytic fragmentation patterns during wounding. Our generated data will function as a starting point for follow-up studies for biomarker studies on larger well-defined patient groups and can be used as a database for many other studies aimed at investigating specific processes during wound healing and infection. Furthermore, it may aid in the discovery of novel bioactive peptides and elucidation of their biological roles.